# Learning Implicit PDE Integration with Linear Implicit Layers

**Marcel Nonnenmacher**
Institute of Coastal Systems, Helmholtz-Zentrum Hereon
Geesthacht, Germany
`marcel.nonnenmacher@hereon.de`

**David S. Greenberg**
Institute of Coastal Systems, Helmholtz-Zentrum Hereon
Geesthacht, Germany
`david.greenberg@hereon.de`

## Abstract

Neural networks can learn local interactions to faithfully reproduce large-scale dynamics in important physical systems. Trained on PDE integrations or noisy observations, these emulators can assimilate data, tune parameters and learn sub-grid process representations. However, implicit integration schemes cannot be expressed as local feedforward computations. We therefore introduce linear implicit layers (LILs), which learn and solve linear systems with locally computed coefficients. LILs use diagonal dominance to ensure parallel solver convergence and support efficient backward mode differentiation. As a challenging test case, we train emulators on semi-implicit integration of 2D shallow-water equations with closed boundaries. LIL networks learned compact representations of the local interactions controlling the 30.000 degrees of freedom of this discretized system of PDEs. This enabled accurate and stable LIL-based emulation over many time steps where feedforward networks failed.

## 1 Introduction

Numerical integration of partial differential equations (PDEs) is essential for many quantitative sciences, including fluid dynamics, numerical weather prediction and climate science [1, 6, 22]. Simulating these systems at scale requires substantial effort and computational resources [8, 36].

Recent studies used machine learning to emulate the PDE integration schemes that define full simulation models [12, 37, 29, 35] or their physical subcomponents [38, 26]. Applications include accelerated integration [37], system identification [18], automatic differentiation for data assimilation [2, 9, 24] and model tuning to match observations [32]. Many PDE integration schemes based on machine learning employ convolutional neural networks (CNNs) [18, 23, 21], whose kernels can precisely express spatial differential operators discretized as translation-invariant stencils.

However, a CNN that can express all of a PDE's mathematical terms may still fail to emulate numerical integration with implicit schemes, as required for stiff PDEs such as incompressible flow [1] and chemical dynamics [30]. Implicit schemes update the state by solving a systems of equations, and while the equation coefficients usually depend on local information, their solutions can depend on the entire previous state (Fig. 1a). This long-range spatial dependence prevents purely feed-forward networks from scaling efficiently to large systems.

35th Conference on Neural Information Processing Systems (NeurIPS 2021), Sydney, Australia.

To address this limitation, we introduce linear implicit layers (LILs), which define and solve a linear system with locally determined coefficients. This operation defines a nonlinear, differentiable implicit function with long-range spatial dependencies. LILs differ fundamentally from existing implicit layers based on Lipschitz constraints [7, 20], quadratic programming [39] or root-finding [5], instead constraining the linear system to be diagonally dominant to ensure convergence with a parallel solver in the forward pass. An efficient backward pass is implemented as a second linear solve. Using a challenging test problem based on the 2D shallow-water equations [34], we show that neural architectures incorporating LILs can learn dynamics integrated with semi-implicit schemes.

## 2 Methods

### 2.1 Problem Statement: Dynamical System Emulation

Let $x_0, x_{\Delta t}, x_{2\Delta t}, \ldots$ be a sequence of system states produced by numerical integration of a space- and time-discretized PDE.[1] Given a collection of such sequences, our aim is to learn the update function $x_{t+\Delta t} = f(x_t)$.

From a data-centric perspective, $f$ predicts the next sequence element. From a PDE-centric view, $f$ numerically integrates over $\Delta t$ time units, with the given discretization and integration schemes. As function family for $f$, we consider deep neural networks consisting of convolutional layers and LILs.

### 2.2 Linear implicit layers

Implicit layers can be written as $F_\theta(y, z) = 0$, where $y$ is the layer input, $\theta$ the parameter vector and and $z$ the layer output [19, 39]. We here consider the particular case of a system of linear equations,

$$A_\theta(y) \, \text{vec}(z) = b_\theta(y). \tag{1}$$

where coefficients $A_\theta, b_\theta$ are local functions of $y$ and are parametrized by some vector $\theta$. While the output $z$ is a solution to a linear system, it depends nonlinearly on the inputs $y$. Firstly, since $A_\theta(y)$ and $b_\theta(y)$ are in general nonlinear functions, and second since $\text{vec}(z) = A_\theta(y)^{-1} b_\theta(y)$ depends bilinearly on $A_\theta(y)^{-1}$ and $b_\theta(y)$. We parameterize $A_\theta$ instead of its inverse to efficiently describe linear constraints $z$ which are sparse and local, which would not be the case for $A_\theta(y)^{-1}$ (Fig. 1a).

When viewed as function $f_\theta(y) = z$, then for use as a network layer, we need to be able to evaluate $f_\theta$ and we need its partial derivatives wrt. $A, b, \theta$ and $y$ for automatic differentiation. For evaluation of $f_\theta$, all we need is a solver for systems of linear equations with matrix $A_\theta$. For computing gradients of a target loss $\mathcal{L}_\theta(y)$ within reverse-mode differentiation, we need to solve another linear system with matrix $A_\theta^\top$, as stated by straight-forward application of the implicit function theorem [17]:

$$A_\theta^\top \frac{\partial \mathcal{L}_\theta}{\partial \text{vec}(b_\theta)} = \frac{\partial \mathcal{L}_\theta}{\partial \text{vec}(z)}, \tag{2}$$

where $\frac{\partial \mathcal{L}}{\partial z}$ is the backpropagating gradient up to $z$. The implicit function theorem also gives $\frac{\partial \mathcal{L}}{\partial A} = -\frac{\partial \mathcal{L}}{\partial b} \text{vec}(z)^\top$, and the remaining partial derivatives for computing $\frac{\partial \mathcal{L}}{\partial y}, \frac{\partial \mathcal{L}}{\partial \theta}$ then are handled entirely by backpropagation. In particular, $\frac{\partial A}{\partial y}, \frac{\partial A}{\partial \theta}$ and $\frac{\partial b}{\partial y}, \frac{\partial b}{\partial \theta}$ depend on our choice for parametrizing the coefficients $A_\theta(y), b_\theta(y)$.

### 2.3 Parametrization of the system of linear equations

For many applications $\dim(x_t)$ can easily reach between tens of thousands and billions [15]. This requires a sparse parametrization of the coefficient matrix $A_\theta$, whose number of entries otherwise scales quadratically with system size. Importantly, $A_\theta$ also has to be full rank for the mapping $f_\theta(y) = z$ to be well-defined. We choose $A_\theta$ to be a banded matrix. To exploit translation equivariance common to PDEs, we compute the bands of $A_\theta(y)$ as convolutional outputs:

$$\text{diag}_{\sigma_c}(A_\theta) = (\mathbf{M}_\theta(y))_c, \; c = 1, \ldots, C, \tag{3}$$

where $\mathbf{M}_\theta(y)$ is the output tensor of a CNN, $c$ is a channel index, and $\text{diag}_\sigma(A)$ indexes the main diagonal ($\sigma = 0$) and offdiagonals ($\sigma \neq 0$) of $A$. We choose the non-zero bands of $A_\theta$ to correspond

---

[1]We focus on regular Cartesian grids, though our approach is not limited to these [10, 14].

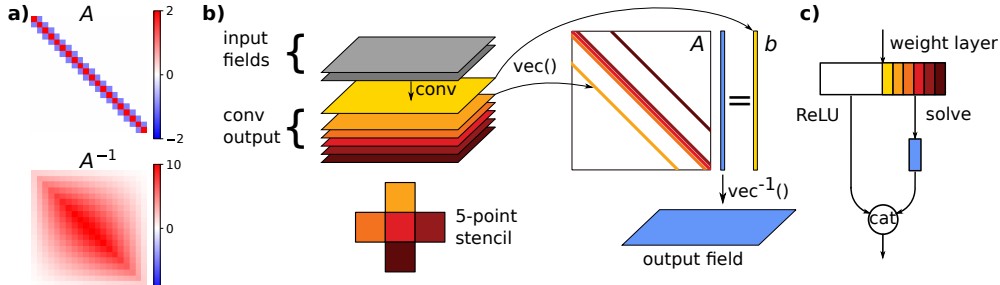

Figure 1: **a)** Sparse banded matrices representing convolutional stencils (upper: second derivative filter [-1,2,-1]) can have dense inverses (lower). **b)** The linear implicit layer generates output $z$ (blue) by solving a linear system $Az = b$, with coefficients computed by convolutional layers applied to the inputs. **c)** Linear implicit layers are straightforward to embed in convolutional networks, by computing $A, b$ from a subset of convolution outputs and concatenating the solution $z$ with the rest.

to elements of computational stencils applied to output $z$: For systems with two spatial dimensions, we select the main diagonal and four off-diagonal bands ($C = 5$) of $A_\theta$ to represent the central and outer points of a classical 5-point stencil, respectively (Fig. 1b). The righthand-side

$$b_\theta = (\mathbf{M}_\theta(y))_{c=C+1} \tag{4}$$

of the system of linear equations is computed as another convolutional output channel. By this choice of $A_\theta(y), b_\theta(y)$, the partial derivatives $\frac{\partial A}{\partial y}, \frac{\partial A}{\partial \theta}, \frac{\partial b}{\partial y}, \frac{\partial b}{\partial \theta}$ are computed from those of the network output $M_\theta(y)$ and, in the case of $\frac{\partial A}{\partial y}, \frac{\partial A}{\partial \theta}$, a simple look-up operation for where the bands are within $A_\theta$. To avoid degeneracies, we normalize each row of $A_\theta$ and $b_\theta$ such that $\text{diag}_0(A_\theta) = 1$.

## 2.4 Linear Solver

The main computational burden of linear implicit layers lies with the solver for the system of equations, which is called twice per gradient step. We use Red-Black Gauss-Seidel [28], an iterative solver that parallelizes well across large system sizes and only requires repeated (sparse) products of the form $A_\theta \, \text{vec}(z)$. This solver requires $A_\theta$ to be diagonally dominant, which we enforce through

$$\text{diag}_0(A_\theta) = \sum_{\sigma \neq 0} |\text{diag}_\sigma(A_\theta)| + \exp((\mathbf{M}_\theta)_k), \tag{5}$$

where $\sigma_k = 0$, before normalizing rows of $A_\theta, b_\theta$. Diagonal dominance ensures $A_\theta$ is full rank.

## 2.5 Boundary conditions

A particular challenge [23] for the definition of the system of equations is posed by boundary conditions (BCs), which enforce equations on the domain boundaries $\Omega$ that can differ drastically from those in the interior. With binary boundary map $I_\Omega$ we compose the full system of equations as

$$\mathbf{M}_{com}(y) = (1 - I_\Omega)\,\mathbf{M}_\theta(y) + I_\Omega\,\mathbf{M}_{BC}(y), \tag{6}$$

and analogous for $b_{com}$. The equations on the boundary are defined by $I_\Omega\,\mathbf{M}_{BC}, I_\Omega\,b_{BC}$, which we can learn with a dedicated model, or in simple cases define manually.

## 3 Numerical experiments on Shallow-Water Equations

We test[2] our implicit integration models on the shallow-water equations (SWE) in two spatial dimensions, with bottom friction and variable depth profile $h$. The SWE describe the interactions of water height $\xi$, vertical and horizontal velocities $v = (v_x, v_y)$. We use a semi-implicit integration

---

[2]Code for reproducing our results is found at `https://github.com/m-dml/lil2021swe`

scheme [4] that advances the system state $x_t = (\xi_t, v_t, h)$ by $\Delta t$ in three steps:

$$v^* = f^*(\xi_t, v_t, h) \qquad \text{interim velocities (explicit)} \qquad (7)$$
$$A^*(v^*)\,\xi_{t+\Delta t} = b^*(v^*) \qquad \text{implicit treatment of water height} \qquad (8)$$
$$v_{t+\Delta t} = f^{\Delta}(\xi_{t+\Delta t}, v^*) \qquad \text{explicit velocity update} \qquad (9)$$

We use an Arakawa C-grid [3] with 100x100 grid points at 10km resolution (Fig. 2a). Depth profile maps $h$ are drawn from correlated Gaussian noise, and BCs enforce $\xi, v_x, v_y = 0$ on the boundary.

Models are trained on 60.000 numerically integrated pairs $(x_t, x_{t+\Delta t})$, with a mean-squared loss to predict the next system state, and data variances normalized per channel. The data consists of numerical simulations from different initial conditions and depth profiles. Initial conditions are $\xi, v_x, v_y = 0$ with different local perturbations to water height $\xi$ of varying strength, spatial extent and location. For each initial condition we simulate 25 hours of dynamics. We use 80% of simulations for training and 20% as validation data. For model training we use Adam [16] with default parameters. We train 5-layer CNNs with 3x3 kernels, both ReLU and bilinear activations [11] and 14.4k free parameters. Six output channels of the 4th convolutional layer are used to define $M_\theta(x_t)$ and $b_\theta(x_t)$ for a 5-point stencil system $A_\theta\,\hat{\xi}_{t+\Delta t} = b_\theta$. $\hat{\xi}_{t+\Delta t}$ is passed to the final network layer to estimate $\hat{v}_{t+\Delta t}$. For comparison, we train fully explicit models (pure CNNs) in which the channels for $M_\theta$ and $b_\theta$ act as regular convolutional channels. We enforce BCs on $\xi$ for the semi-implicit models at the level of $M_{com}, b_{com}$, and for the explicit models and $v$ we enforce them directly on the model output through masking with $I_\Omega$.

We train the models on three datasets with different integration step-sizes $\Delta t \in \{300s, 900s, 1500s\}$. All experiments combined required 160 hours on a V100 GPU. For evaluation we compare trained models to numerical integration on novel initial conditions. Models with LILs perform well for all step sizes, while pure CNNs fail at high $\Delta t$ (Fig. 2b-c), for which long-range dependencies emerge.

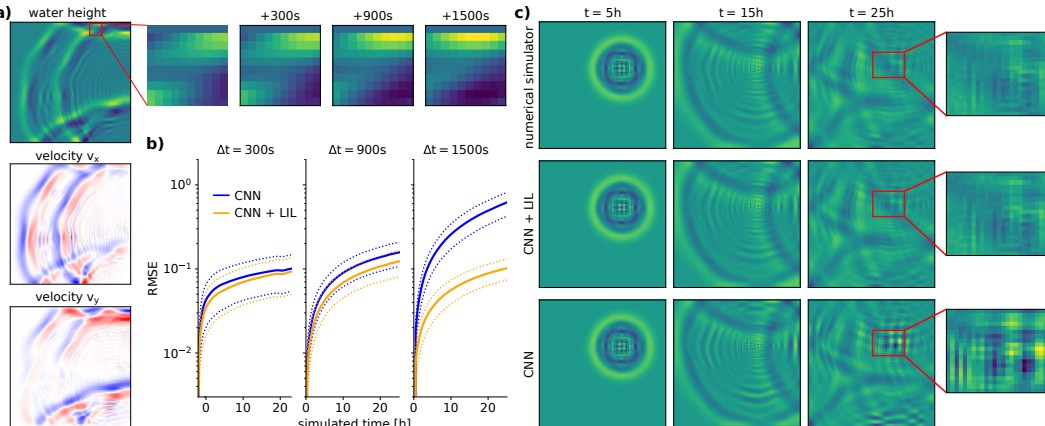

Figure 2: **a**) Shallow water equation system state (left) and magnified surface height dynamics (right). **b**) Mismatch between numerical integration and emulation with CNNs with and without LILs vs. simulated time, for different $\Delta t$. RMSE $\pm$1std over 100 initial conditions. **c**) Comparison of CNNs with and without LILs over 20h with $\Delta t = 1500s$, starting from the same state 5h after initial perturbation.

## 4  Discussion

We showed that LILs enable deep learning of spatially extended dynamics that require semi-implicit integration. Previously Look et al. [19] studied implicitly-integrated neural models on ordinary differential equations with up to 150-dimensional system states. Here we focused on scalability in PDE systems with multiple spatial axes and variable fields, reaching 10.000 grid points with 3 variables each. Choosing diagonally-dominant banded matrices allows fast parallel solvers to scale to large systems, but also limits the expressiveness of LILs. For instance, LILs could learn a fully implicit model scheme for the SWE by defining a system of equations over all system variables,

but with larger systems the requirement of diagonal dominance becomes increasingly restricting, especially for large time steps.

Other LIL parametrizations such as low-rank perturbations of diagonal matrices are possible, as would be other solvers such as conjugate gradient methods [31]. While we used LILs to incorporate the long-range input output dependence appearing in (semi-)implicit schemes, other approaches are possible as well, such as U-Nets [27, 35] and self-attention [33]. Unlike LILs, U-Nets have a restricted dependency range by construction, and self-attention for multi-axis systems typically is spatially restricted as well [25, 13].

LILs provide efficient and straightforward backward-mode differentiation, and require relatively few parameters, requiring only six convolutional channels per output field with a 5-point stencil. Several LILs can be stacked within a network.These features suggest that LILs could prove useful as general building blocks for deep models requiring non-local computations.

## 5    Acknowledgments

M. Nonnenmacher and D. Greenberg were supported by the Helmholtz AI initiative. We thank Kai Logemann for help with the SWE numerical simulator, and Jan-Matthis Lueckmann, Tobias Machnitzki, Shivani Sharma and Vadim Zinchenko for comments on the manuscript.

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
