# OpenReview forum: "Learning Implicit PDE Integration with Linear Implicit Layers"
_NeurIPS.cc/2021/Workshop/DLDE — DLDE Workshop -- NeurIPS 2021 Spotlight_

### Official Review · Reviewer_78bN · 2021-10-03
**Interesting choice of implicit layer to faster optimization**

**Confidence:** 3

**Review:**

### Summary

The article proposes an implicit layer that can take advantage of RBGS solvers to speed up optimization.

### Comments

**pros**: The article is clear and concise. Results for large \Delta t seem promising.

**cons**:

Eq 1: The equation can be rewritten as z = B(y), with a quadratic interaction at the output of B. What do we gain by keeping the equation as presented?

No major weaknesses found in the argumentation.

**Score:**

4: Very good paper

---

### Official Review · Reviewer_BXKb · 2021-10-11
**A computationally efficient framework is proposed to address implicit integration in PDEs. The framework is evaluated on shallow-water equations in two spatial dimensions, showing interesting potential applications.**

**Confidence:** 3

**Review:**

Pros: The paper addresses the important open problem of scaling integration in PDEs to large scale systems. Linear Implicit Layers with diagonally dominant constraints are demonstrated on a system of 30000 variables. The paper is well written, giving a detailed description of the proposed framework and the experimental results.

Cons: Details on the theoretical implications of limiting layers to diagonally dominant matrix representations are needed. What practical system limitations are implied? What next implementation and evaluation steps are proposed?

This paper would make an interesting contribution to the workshop.


**Score:**

4: Very good paper

---

### Official Review · Reviewer_ngTu · 2021-10-11

**Confidence:** 3

**Review:**

The authors propose a method of learning implicit PDE integration, by using linear implicit layers.

The paper is well motivated. However, there are multiple typos and at times it is hard to follow the paper, and some of the details are not very clear.
Anyhow, I am convinced that the idea and motivation is correct and sound, and this paper will be a good contribution for the workshop, and perhaps in this context more details and unclarities could be discussed and shared.



**Score:**

3: Good paper

---

### Official Review · Reviewer_TKXj · 2021-10-11
**A fair contribution in the use of an implicit layer in a CNN to better emulate implicit PDE integrators**

**Confidence:** 1

**Review:**

The paper proposes a method using CNNs to emulate an implicit PDE numerical solver.  This is a challenge since typical CNNs in the literature have small support kernels that do not model longer range interactions in implicit solvers for PDE.  This causes inefficiency of the CNN to learn the implicit solver.  The proposed approach embeds an implicit layer that is the solution of a parameterized linear system into the CNN.  The derivative computation for back-prop through this implicit layer is shown.  The solver proposed for the forward/backward pass is the red-black Gauss-Seidel method.  Example application is given on the Shallow-Water Eqns; a 5-layer CNN is constructed with the 4th layer using the linear implicit layer (LIL).  Comparison of the standard CNN and the CNN+LIL is shown.   CNN-LIL has lower error compared to the standard integrator.

I find the use of implicit layers in this application of numerical integration of a PDE a fair contribution, and the proof-of-concept experiments are given to validate the idea.  As the topic also fits well into this workshop, I think it should be accepted though I really don’t know well the literature on CNN emulation of PDEs.

Comments:
1) Why is only 1 layer an LIL in the proposed CNN?  Can you use the embedding in other layers?
2) What is the speed emulation compared to standard PDE integrator?
3) What are the initial conditions for that 60k training set (Ln 95-96 is unclear)?  Better discussion of training data would be good.
4) Have you tried other linear solvers (e.g., conjugate gradient; multigrid)?  What is the time comparison?
5) What is the inference time comparison of CNN vs CNN-LIL?
6) Provide more details on why your approach is advantageous over the U-Net approach mentioned in the Sect 4.

**Score:**

3: Good paper

---

### Decision · Program_Chairs · 2021-10-15

**Decision:**

Accept (Spotlight)

**Comment:**

The submission received very good reviews, with some minor concerns regarding clarity. The reviewers were eager to hear more, and so this submission would make a good spotlight talk for the workshop.

Although not noted by the reviewers, I would recommend clarification regarding the claim of the method working on a system of 30,000 variables. It seems this number refers to the size of the vector used to discretize the solution fields over the (2+1 dimensional?) problem domain. Given that neural networks are also studied to solve PDEs posed in high-dimensional domains, it is worth being more specific about the exact nature of this number 30,000 (where it comes from, what it means, how it affects computational cost of competing methods, etc.).